# Dissociation Between Tumor Response and PTTM Progression During Entrectinib Therapy in NTRK Fusion-Positive Colon Cancer

**DOI:** 10.3390/curroncol32090506

**Published:** 2025-09-11

**Authors:** Hideki Nagano, Shigekazu Ohyama, Atsushi Sato, Jun Igarashi, Tomoko Yamamoto, Mikiko Kobayashi

**Affiliations:** 1Department of Surgery, Marunouchi Hospital, Matsumoto 390-8601, Nagano, Japan; 2Department of Pathology, Marunouchi Hospital, Matsumoto 390-8601, Nagano, Japan

**Keywords:** pulmonary tumor thrombotic microangiopathy, neurotrophic tropomyosin receptor kinase gene fusion, pulmonary hypertension, acute respiratory failure, right heart failure, entrectinib, transverse colon cancer

## Abstract

We report a rare case of pulmonary tumor thrombotic microangiopathy (PTTM) in a patient with neurotrophic tropomyosin receptor kinase (NTRK) fusion-positive colorectal cancer (CRC) who exhibited marked tumor regression with entrectinib. Despite significant tumor shrinkage, the patient developed fatal respiratory failure, and an autopsy revealed tumor emboli and pulmonary vascular remodeling. In this case, PTTM developed via an atypical lymphatic route through the thoracic duct, without hematogenous metastasis. Notably, the tumor showed microsatellite stability and only a modest mutation burden, suggesting that lymphatic spread and microvascular pathology may progress independently of these genomic features. This case underscores a critical dissociation between oncologic response and vascular complications, indicating that tropomyosin receptor kinase (TRK) inhibitor monotherapy may be insufficient in preventing PTTM. Comprehensive management may require concurrent strategies targeting the pulmonary microvasculature, including antiangiogenic therapy and modulation of cytokine and growth factor signaling.

## 1. Introduction

Pulmonary tumor thrombotic microangiopathy (PTTM) is a rare but aggressive pulmonary vascular complication of malignancies. It is characterized by tumor embolization in small pulmonary arteries, leading to acute respiratory failure and pulmonary hypertension (PH). Obstruction of the pulmonary microvasculature triggers fibrocellular intimal proliferation and activation of the coagulation cascade, resulting in rapid clinical deterioration and high mortality rates [1,2,3]. Due to its nonspecific imaging features and fulminant progression, PTTM is often diagnosed postmortem [2,3]. Patients typically present with dyspnea, hypoxia, thrombocytopenia, and signs of a hypercoagulable state. In a large autopsy series of 2215 cancer patients, PTTM was identified in 30 cases (1.4%), with gastric adenocarcinoma being the most common primary tumor [3]. Notably, none of the 30 PTTM cases involved colorectal cancer (CRC), suggesting that PTTM is exceptionally rare in CRC. However, emerging case reports indicate that it may occur even in molecularly defined subtypes such as NTRK fusion-positive tumors. To our knowledge, no prior reports have described PTTM in NTRK fusion-positive CRC.

CRC ranks as the third most commonly diagnosed malignancy worldwide and remains the second leading cause of cancer-related mortality [4]. While conventional chemotherapy forms the backbone of treatment, rare molecular subtypes—such as neurotrophic tropomyosin receptor kinase (NTRK) gene fusion-positive CRC—require distinct therapeutic approaches [5]. NTRK gene fusion leads to constitutive activation of tropomyosin receptor kinase (TRK) signaling, driving oncogenesis via the MAPK and PI3K pathways [6,7]. Although TRK inhibitors such as larotrectinib and entrectinib have demonstrated efficacy in treating NTRK fusion-positive CRC, the mechanisms underlying disease progression—including the development of PTTM—remain poorly understood. Specifically, we present a case of NTRK fusion-positive CRC in which PTTM developed despite a marked tumor response to entrectinib. This case offers insight into a potential disconnect between molecular tumor control and vascular disease progression. The development of PTTM, despite entrectinib’s antitumor efficacy, suggests that oncogenic signaling and pulmonary vascular pathology may proceed through partially independent mechanisms, underscoring the need for integrated therapeutic strategies and heightened clinical vigilance.

## 2. Case Presentation

A 70-year-old Japanese male with a history of smoking and alcohol consumption underwent an annual health check-up, during which a fecal occult blood test returned positive for the first time. Subsequent investigations led to a diagnosis of transverse colon cancer with peritoneal metastases. Contrast-enhanced computed tomography (CECT) revealed an irregular mass in the transverse colon near the hepatic flexure, along with multiple peritoneal nodules primarily localized to the greater omentum (Figure 1A). Chest CT revealed multiple blebs in both lung apices and a reticular pattern in the right lung (Figure 1B). Despite peritoneal dissemination, no hematogenous or lymphatic metastases were evident at this stage.

Initial laboratory findings revealed elevated tumor markers, including carcinoembryonic antigen (CEA) at 55.0 ng/mL and carbohydrate antigen 19-9 (CA19-9) at 2527.6 U/mL. Genetic analysis confirmed that the tumor was RAS/BRAF wild-type and microsatellite instability (MSI)-negative. The patient had a history of emphysema but did not require supplemental oxygen therapy.

The initial treatment regimen consisted of capecitabine, oxaliplatin, and cetuximab. After three cycles, CT imaging revealed disease progression, with the transverse colon tumor increasing from 40 × 34 mm to 44 × 36 mm and peritoneal nodules enlarging from 44 × 32 mm to 52 × 35 mm. The regimen was subsequently modified to capecitabine, irinotecan, and bevacizumab. Despite eight cycles, the disease progressed, with marked ascites accumulation and mesenteric shortening. CT at this stage showed poorly defined masses in both the transverse colon and peritoneal surfaces, without evidence of hematogenous metastases (Figure 2A). Compared with pretreatment findings, chest CT revealed no significant worsening of reticular or interstitial shadows (Figure 2B).

A third-line regimen of trifluridine/tipiracil (FTD/TPI) plus bevacizumab was initiated. After three cycles, ascites further increased, necessitating cell-free and concentrated ascites reinfusion therapy (CART). Throughout this period, the patient’s mild exertional dyspnea, attributable to underlying pulmonary emphysema, remained unchanged, and no respiratory symptoms suggestive of PTTM were observed. Given the need for alternative therapeutic strategies, a genetic panel analysis was performed, revealing an NTRK1-LMNA gene fusion. Based on the established efficacy of TRK inhibitors in NTRK fusion-positive malignancies, entrectinib treatment was promptly initiated.

Within six weeks of entrectinib treatment, the patient exhibited a remarkable response. Tumor marker levels decreased significantly, with CA19-9 levels dropping from >24,000 U/mL (assay upper limit) to 2470.2 U/mL. Clinically, ascites accumulation stabilized, and three rounds of CART were performed to manage symptoms. Throughout this period, the patient remained relatively stable and continued outpatient care.

Two months after initiating entrectinib therapy, the patient developed sudden dyspnea, fever, and hypoxia, necessitating emergency hospitalization. Upon transport, his peripheral oxygen saturation was critically low at 50% on room air, improving only marginally to the low 80 s with 15 L/min oxygen. On admission, vital signs included a blood pressure of 102/56 mmHg, a pulse rate of 80 beats per minute, a respiratory rate of 20 breaths per minute, and a temperature of 38.5 °C.

High-flow oxygen therapy was initiated; however, oxygen saturation remained unstable. Laboratory findings revealed worsening anemia, elevated liver enzymes, and markedly increased C-reactive protein (CRP) levels, consistent with a systemic inflammatory response. Chest CT imaging revealed marked progression of interstitial pneumonia in both lungs, most prominently in the right lobe (Figure 3A). Newly developed ground-glass opacities appeared in previously unaffected lung regions amid areas of emphysema, accompanied by multiple infiltrative shadows, mild right lung volume reduction, and bilateral pleural effusion (Figure 3B). These findings raised suspicion for exacerbation of interstitial pneumonia, with possible superimposed bacterial infection or drug-induced lung injury. Cardiogenic pulmonary edema and pulmonary embolism were also considered as differential diagnoses. However, due to the patient’s poor general condition on the day of admission, neither echocardiography nor contrast-enhanced CT could be performed, limiting further evaluation. Although PTTM was not initially suspected, its possibility could not be excluded at this stage. Bronchoalveolar lavage and other invasive diagnostic procedures were also deferred due to compromised respiratory status. CT imaging of the chest and abdomen revealed no evidence of hematogenous metastasis, such as lung or liver involvement.

By the third day of hospitalization, signs of right-sided heart failure, including jugular vein distention and peripheral edema, had become apparent. Echocardiography revealed a peak tricuspid regurgitation velocity (TRV) of 4.5 m/s, a tricuspid regurgitation pressure gradient (TRPG) of 82.8 mmHg, and an estimated systolic pulmonary artery pressure (SPAP) of 86 mmHg, consistent with severe pulmonary hypertension. Given the suspected presence of tumor emboli within pulmonary vessels, PTTM was considered a potential underlying cause.

Despite discontinuation of entrectinib—due to the inability to rule out drug-induced lung injury—and initiation of empirical therapy with meropenem and methylprednisolone, the patient’s condition continued to deteriorate. Worsening pleural effusion necessitated bilateral thoracentesis for symptom relief. As pulmonary function deteriorated, enterctinib was reintroduced based on the hypothesis that TRK-driven tumor embolization within the pulmonary vasculature was contributing to disease progression. This decision was supported by the lack of hematogenous metastases, which strengthened the suspicion of intravascular tumor spread rather than conventional metastatic dissemination, and by the absence of clinical improvement despite anti-infective and anti-inflammatory treatment, with no viable alternative systemic therapies available.

Despite these efforts, the patient’s respiratory distress worsened, and he passed away seven weeks after hospitalization. An autopsy was performed with the family’s consent.

### 2.1. Autopsy Findings

#### 2.1.1. Macroscopic Findings

In the right thoracic cavity, minor adhesions were observed alongside 500 mL of clear yellowish pleural effusion. In contrast, the left thoracic cavity exhibited severe adhesions, hindering lung removal, with almost no pleural effusion present. The abdominal cavity displayed extensive fibrous adhesions, with diffuse white fibrous tissue uniformly covering the serosal surfaces of the intestinal tract, resulting in firm intersegmental adhesions (Figure 4A). No gross tumor deposits were identified on the serosal surfaces, and ascites was minimal. The primary transverse colon lesion had significantly regressed, leaving a 3 cm square mucosal area containing several polyps, the largest measuring up to 1 cm (Figure 4B). No additional mass lesions, including disseminated foci, were detected in the abdominal cavity.

Prominent lymph node enlargement was noted in the abdominal cavity and along the periaorta. The lungs exhibited diffuse consolidation with areas of sclerosis, along with apical bullae and microcystic changes consistent with smoking-related emphysematous alterations (Figure 4C). No metastatic mass lesions were found within the lung parenchyma; however, multiple hilar lymphadenopathies were observed. The heart, liver, and kidneys showed no significant abnormalities.

#### 2.1.2. Microscopic Findings

At the primary transverse colon site, chemotherapy demonstrated significant efficacy, with well-to moderately differentiated tubular adenocarcinomas present in several polypoid areas within the mucosal layer. Scattered microscopic remnants were identified in the submucosal layer, although no evidence of lymphatic or venous invasion was observed (Figure 5A). Adenocarcinoma with extensive fibrosis was diffusely spread across the intestinal surface, resulting in adherence and direct invasion of adjacent organs including the stomach, liver, bladder, and diaphragm (Figure 5B). Additionally, widespread dissemination of adenocarcinoma was noted in the left thoracic cavity. Tumor emboli were identified within lymphatic vessels across multiple organs, including the stomach, bladder, and pleura. Regarding lymph node metastasis, multiple metastatic lesions were present in the abdominal cavity, pulmonary hilum, and para-aortic region.

In the lungs, diffuse bronchopneumonia and organizing pneumonia were observed (Figure 5C,D), likely resulting from aspiration pneumonia, although drug-related effects contributing to organizing pneumonia could not be entirely excluded. Histopathological findings consistent with PTTM were identified in both lungs, characterized by tumor emboli, fibrous intimal thickening, and thrombus formation within numerous small pulmonary arteries (Figure 5E,F). Additionally, a small number of embolized lymphatic vessels were observed. No hematogenous metastases were identified in the lung parenchyma, highlighting the distinct vascular involvement characteristic of PTTM. The absence of hematogenous metastases, despite extensive pulmonary vascular pathology, supports the hypothesis that PTTM may have developed through lymphatic dissemination rather than direct hematogenous spread. Furthermore, the presence of tumor emboli within lymphatic vessels across multiple organs—including the pleura, stomach, and bladder—reinforces this hypothesis. These findings suggest that lymphatic migration of tumor cells may have contributed to pulmonary microvascular obstruction and intimal proliferation, ultimately leading to the development of PTTM.

Table 1 provides an integrated overview of the autopsy findings by organ system, including both gross and microscopic observations, with particular attention to metastatic involvement and PTTM-related features.

The tumor mutational burden (TMB) was 1.26 mut/Mb, indicating a relatively low level of genetic alterations.

A definitive postmortem diagnosis of PTTM was established, suggesting that the condition led to severe PH and right heart failure, progressing rapidly over a short period.

## 3. Discussion

To our knowledge, this is the first reported case of PTTM occurring in a patient with NTRK fusion-positive CRC treated with entrectinib. This distinguishes the present report from prior descriptions of PTTM in gastrointestinal malignancies, primarily involving gastric adenocarcinoma and often lacking molecular characterization. PTTM is a rare but fatal complication of malignancy, characterized by fibrocellular intimal proliferation induced by tumor microemboli, vascular remodeling, and progressive PH [8]. Unlike conventional pulmonary embolism, PTTM is driven by cytokine-mediated vascular remodeling, particularly involving platelet-derived growth factor (PDGF) and vascular endothelial growth factor (VEGF), which trigger endothelial activation, intimal thickening, and microvascular obstruction [2]. Chronic thromboembolic pulmonary hypertension (CTEPH), a sequela of unresolved embolism, may also exhibit vascular remodeling mediated by similar inflammatory and angiogenic factors—including PDGF, VEGF, and TGF-ß—suggesting partial mechanistic overlap. However, PTTM is uniquely characterized by intravascular tumor cell invasion and abrupt mediator release, resulting in rapid and aggressive microvascular obstruction. Therefore, despite overlapping molecular pathways, its clinical progression is markedly more aggressive than that of CTEPH [3,9]. The diagnosis of PTTM is exceptionally difficult during life due to its rapid progression and nonspecific clinical and radiographic findings. Ground-glass opacities, interstitial lung patterns, and pleural effusion can mimic chemotherapy-induced pneumonitis or infection, often delaying recognition and treatment. Therefore, in patients with malignancies who present with sudden respiratory failure and PH during systemic therapy, a high index of suspicion for PTTM is warranted.

The patient demonstrated resistance to conventional chemotherapies, including cetuximab and bevacizumab; however, entrectinib showed remarkable antitumor activity. Notably, despite effective entrectinib treatment, the patient developed sudden-onset acute respiratory failure and right heart dysfunction due to PH. PTTM was confirmed postmortem and proved rapidly fatal. This case underscores the importance of recognizing PTTM in patients with malignancies who develop acute respiratory failure during systemic therapy. These findings highlight the urgent need for early recognition and comprehensive therapeutic strategies that extend beyond tumor control to address cancer-associated vascular complications and improve patient outcomes.

Notably, no hematogenous metastases were identified in this case, yet prominent lymphovascular invasion suggested a possible route of dissemination via the thoracic duct, bypassing conventional venous circulation. Given the rarity of NTRK fusion-positive CRC, its metastatic behavior remains poorly characterized. Emerging reports have described a tendency toward lymphatic spread in this molecular subtype, potentially contributing to the pathogenesis of PTTM and warranting further investigation into alternative mechanisms of tumor embolization [10]. Prior studies have documented frequent lymph node metastases in NTRK fusion-positive CRC, supporting the plausibility of lymphatic dissemination as a driver of vascular involvement [11,12]. While this subtype is often associated with microsatellite instability and high tumor mutation burden, the present case exhibited microsatellite stability and only a modest mutation burden, suggesting that lymphatic spread and microvascular pathology may occur independently of these genomic features. These findings underscore the importance of considering noncanonical pathways of disease progression in molecularly defined CRC subtypes, particularly in the context of atypical vascular complication such as PTTM.

NTRK gene fusions result in constitutive activation of TRK receptors and serve as oncogenic drivers across multiple cancer types [13]. In CRC, LMNA:NTRK1 and TPM3:NTRK1 are among the most common fusion partners [13]. These tumors are typically RAS/BRAF wild-type and resistant to anti-EGFR therapies [14]. While frequently associated with MSI-high and high TMB, a subset of NTRK fusion-positive CRCs—such as in the present case—may be microsatellite stable and exhibits a modest TMB, reflecting molecular heterogeneity within this rare group [14,15].

Despite the dramatic tumor regression achieved with entrectinib, PTTM progressed rapidly, ultimately resulting in a fatal outcome. A similar paradoxical case has been reported in advanced non-small cell lung cancer, where fatal PTTM occurred despite near-complete tumor regression following immune-chemotherapy [10]. These observations reinforce that tumor shrinkage does not necessarily equate to vascular disease control. This dissociation highlights the need for clinicians to remain vigilant for microangiopathic complications, even when systemic therapy appears effective. Addressing cancer-associated vascular remodeling requires therapeutic strategies that extend beyond cytoreduction. This dissociation between tumor control and pulmonary vascular complications underscores that effective antitumor therapy alone may be insufficient in managing malignancies complicated by systemic microangiopathic processes. Given the involvement of PDGF and VEGF in PTTM pathogenesis, antiangiogenic therapeutic agents—such as imatinib and bevacizumab—may offer potential benefits [16]. The vascular nature of PTTM suggests that combining TRK inhibitors with antiangiogenic agents could represent a rational strategy. However, this approach remains speculative, as no clinical evidence currently supports its efficacy in NTRK fusion-positive CRC complicated by PTTM. Further investigation is warranted to determine whether vascular-targeted therapies can mitigate microangiopathic progression in this setting. Additionally, targeted agents previously considered ineffective in conventional treatment paradigms may hold therapeutic value when incorporated into a multimodal framework tailored to both tumor biology and vascular involvement.

## 4. Conclusions

This case demonstrates that PTTM may develop despite marked tumor regression achieved through targeted therapy, underscoring a critical dissociation between tumor control and vascular complications. Effective management may require integrated strategies that also target the pulmonary microvasculature. Future research should focus on identifying predictive biomarkers and evaluating combined approaches, including TRK inhibition and antiangiogenic therapy, to improve outcomes in patients with molecularly defined CRC, particularly those at risk for vascular complications such as PTTM.

## Figures and Tables

**Figure 1 curroncol-32-00506-f001:**
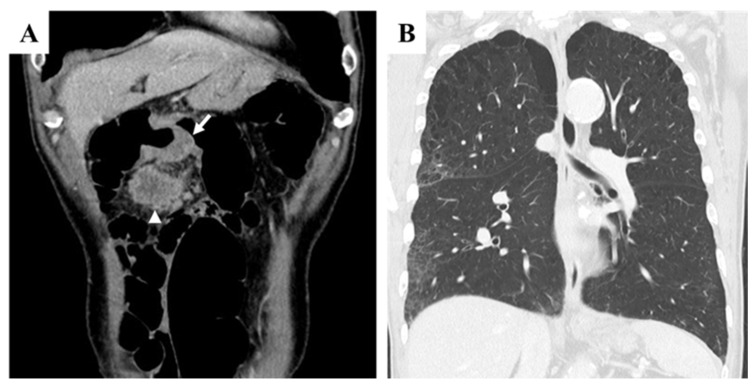
Pretreatment findings from abdominal contrast-enhanced CT and chest CT. (**A**): Coronal section view showing a tumor in the transverse colon near the hepatic flexure (arrow) and a peritoneal metastatic nodule located just caudal to the primary tumor (arrowhead). Several additional peritoneal metastatic nodules were observed along the greater omentum. (**B**): Chest CT revealed multiple blebs in both lung apices, along with a reticular pattern in the right lung lobes.

**Figure 2 curroncol-32-00506-f002:**
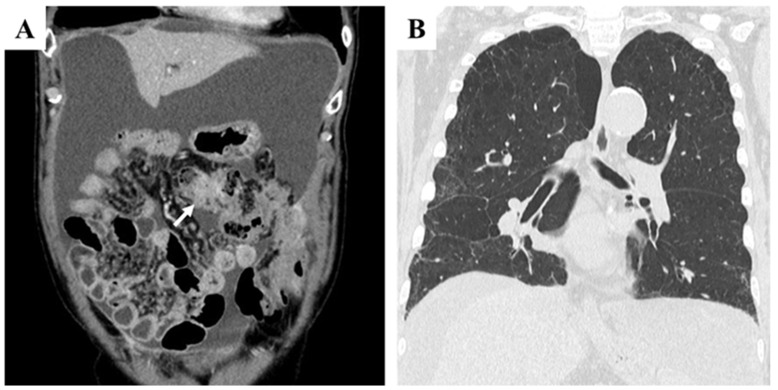
Abdominal contrast-enhanced CT (CECT) and chest CT findings after eight cycles of chemotherapy with capecitabine, irinotecan, and bevacizumab, following resistance to capecitabine, oxaliplatin, and cetuximab. (**A**): Coronal section view showing a large volume of ascitic fluid and mesenteric shortening. The tumor in the transverse colon appeared as an irregular mass (arrow). (**B**): Chest CT revealed no significant progression of reticular or interstitial shadows compared with pretreatment findings.

**Figure 3 curroncol-32-00506-f003:**
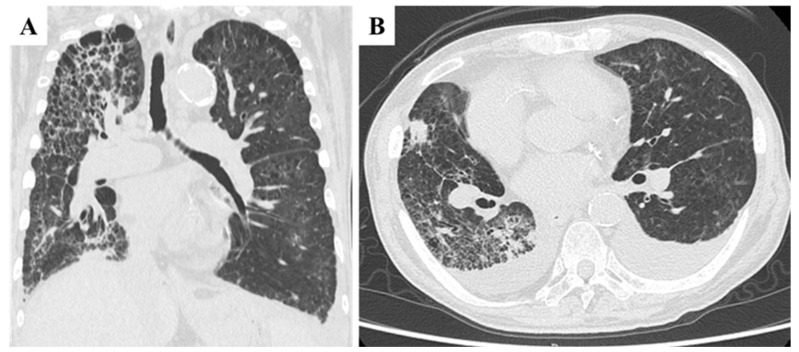
Chest CT findings at the time of transport due to acute respiratory failure. (**A**,**B**): Marked deterioration of interstitial pneumonia was observed in both lobes, with the most pronounced changes in the right lobe. Ground-glass opacities appeared against a background of emphysematous lungs, accompanied by multiple infiltrative shadows, mild right lung volume reduction (**A**), and bilateral pleural effusion (**B**).

**Figure 4 curroncol-32-00506-f004:**
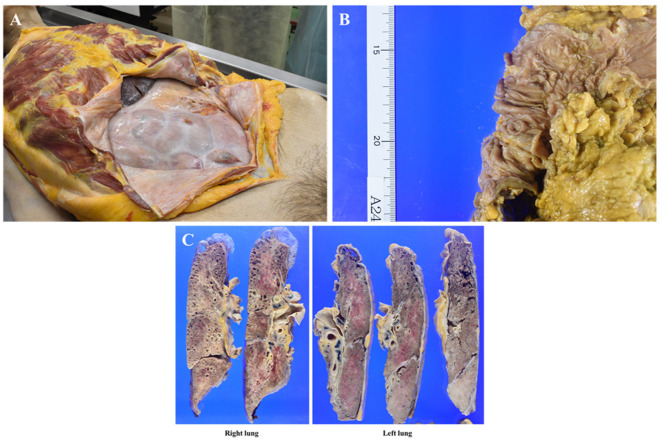
Macroscopic autopsy findings. (**A**): The intestinal tract is uniformly covered with white fibrous tissue, leading to extensive adhesion between structures. (**B**): The primary lesion in the transverse colon has markedly decreased in size following effective treatment, appearing as clustered polyps within a 3 cm square area, the largest measuring up to 1 cm. (**C**): Cross-sections of both lungs revealing diffuse consolidation, apical bullae, and multiple small cysts. The left lung was adherent to the parietal pleura, with enlargement of both hilar lymph nodes.

**Figure 5 curroncol-32-00506-f005:**
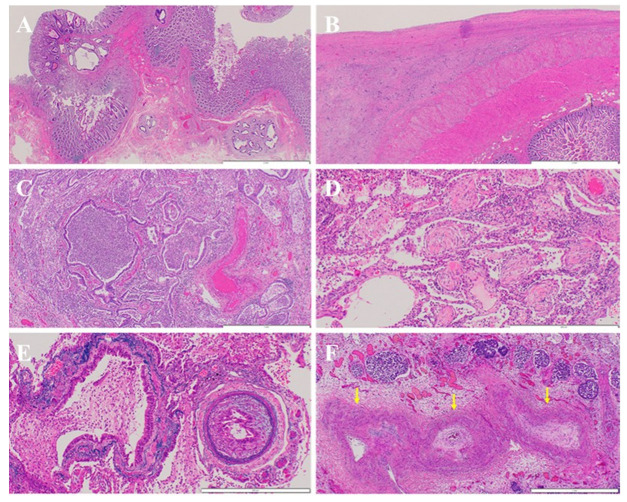
Microscopic findings at autopsy. (**A**): At the primary site in the transverse colon, chemotherapy was highly effective, with well- to moderately differentiated adenocarcinomas observed in several polypoid regions within the mucosal layer. Microscopically, scattered remnants were detected in the submucosal layer, but no evidence of lymphatic or venous invasion was found (white bar at bottom right: 2 mm). (**B**): Adenocarcinoma with severe fibrosis diffusely infiltrated the intestinal tract, leading to adhesion to its surface (white bar at bottom right: 2 mm). (**C**,**D**): In the lungs, diffuse bronchopneumonia (**C**) and organizing pneumonia (**D**) were observed. (**E**,**F**): Findings consistent with PTTM were identified. A small pulmonary artery was embolized by adenocarcinoma ((**E**); Victoria blue and hematoxylin-eosin staining), whereas another pulmonary artery exhibited a narrowed lumen due to intimal thickening ((**F**); arrows) (**E**): white bar at bottom right: 500 μm; (**F**): white bar at bottom right: 1 mm).

**Table 1 curroncol-32-00506-t001:** Summary of Autopsy Findings by Organ System.

Organ/	Gross Findings	Microscopic	Meta	PTTM
System		Findings	Stasis	Features
Lungs	Diffuse consolidation,	Bronchopneumonia,	No	Present
	apical bullae,	organizing pneumonia,		
	microcystic changes,	tumor emboli in small		
	pleural effusion	arteries		
Colon	3 cm mucosal lesion	Residual adenocarcinoma	Yes	Absent
(primary)	with polyps, no serosal	in mucosa/submucosa,	(LN)	
	masses	no vascular invasion		
Abdominal	Extensive fibrous	Diffuse fibrosis with	Yes	Absent
Cavity	adhesions, minimal	tumor spread to stomach,	(LN)	
	ascites	liver, bladder, diaphragm		
Lymph	Enlarged nodes in	Multiple metastaic	Yes	-
Nodes	abdomen, periaorta,	foci		
	pulmonary hilum			
Heart, Liver,	No significant gross	No notable pathology	No	Absent
Kidney	abnormalities			
Pleura	Effusion (right)	tumor emboli in lymphatics	Yes	-
	adhesions (left)			

## Data Availability

The dataset supporting the conclusion of this article is included within the article.

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
