# Peer review of "Dissociation Between Tumor Response and PTTM Progression During Entrectinib Therapy in NTRK Fusion-Positive Colon Cancer"

_curroncol, 2025, doi:10.3390/curroncol32090506_

Round 1
Reviewer 1 Report
Comments and Suggestions for Authors
This case report describes a rare and clinically significant presentation of pulmonary tumor thrombotic microangiopathy (PTTM) in a patient with metastatic NTRK fusion-positive transverse colon cancer who demonstrated a marked radiologic and biochemical response to entrectinib. The manuscript is well-structured, with a clear chronology of clinical events, detailed autopsy findings, and an insightful discussion linking the vascular pathophysiology of PTTM to potential therapeutic strategies. The topic is timely and relevant, as it highlights a lethal complication that may occur despite effective targeted therapy, underscoring the need for multidisciplinary management approaches. Overall, the report provides valuable clinical and pathological insights; however, certain sections could be strengthened by streamlining background information, clarifying diagnostic reasoning, and integrating additional literature context to enhance the manuscript’s impact.
Title & Abstract
- The title is informative and specifies disease, genetic alteration, treatment, and context. However, it is long and may be more concise for better readability (e.g., consider removing redundant qualifiers such as “effective” before entrectinib, as the response is clarified in the abstract).
- The abstract is clear and summarizes the key findings, but the final sentences repeat ideas stated earlier. Consider tightening the discussion of combined strategies to avoid redundancy. Adding more explicit data on the timing between entrectinib initiation and onset of PTTM could strengthen the clinical relevance.
Introduction
- Clarifying how frequently PTTM occurs in colorectal cancer (currently implied as rare, but incidence data, even from autopsy series, would help).
- A more explicit rationale for reporting this case (e.g., novelty due to occurrence despite tumor shrinkage with TRK inhibition).
Case Presentation
- Suggest summarizing earlier chemotherapy regimens more concisely and focusing on changes in disease burden and symptoms that relate to PTTM risk.
- Lines 116–123: The interpretation of CT findings as interstitial pneumonia vs. drug toxicity vs. infection is appropriate, but authors could indicate if bronchoalveolar lavage or other diagnostics were considered.
- Lines 137–143: The rationale for reintroducing entrectinib after discontinuation could be expanded—particularly given potential drug-induced lung injury.
Autopsy Findings
- Consider including a brief table summarizing organ involvement, presence/absence of metastases, and PTTM features for quick reference.
- Lines 186–189: The emphasis that no hematogenous metastases were found strengthens the PTTM mechanism discussion, but authors could relate this more directly to lymphatic spread hypotheses.
Discussion
- Lines 230–235: The hypothesis of thoracic duct-mediated spread is interesting, this would be strengthened by citing more literature on lymphatic spread in NTRK-positive CRC.
- Lines 243–249: The suggestion of combining TRK inhibitors with antiangiogenic therapy is logical but speculative; acknowledging lack of clinical evidence in CRC-PTTM cases would provide balance.
- A brief comparison to similar published cases where tumor control did not prevent PTTM would enhance the clinical learning points.
Conclusion
- Lines 252–258: The conclusion is appropriate but could be more concise, emphasizing the main takeaway which is PTTM can develop despite strong tumor responses and needs parallel vascular-targeted approaches.
Author Response
We sincerely appreciate your thoughtful and constructive comments on our manuscript. In accordance with your suggestions, we have revised the text to address each point raised. Below, we provide detailed explanations of the changes made in response to your feedback. Page and line numbers refer to the revised manuscript with tracked changes, and the modified text is highlighted in red clarity.
Title & Abstract
- The title is informative and specifies disease, genetic alteration, treatment, and context. However, it is long and may be more concise for better readability.
→  Thank you for your valuable suggestion regarding the title. In accordance with your comment, we have revised the title to be more concise while emphasizing the key clinical insight of this case—the dissociation between tumor response and PTTM progression. The revised title is: “Dissociation Between Tumor Response and PTTM Progression During Entrectinib Therapy in NTRK Fusion-Positive Colon Cancer.”
- The abstract is clear and summarizes the key findings, but the final sentences repeat ideas stated earlier. Consider tightening the discussion of combined strategies to avoid redundancy. Adding more explicit data on the timing between entrectinib initiation and onset of PTTM could strengthen the clinical relevance.
→ Thank you for your insightful comments regarding the abstract. In response, we have revised the final sentences to avoid redundancy and to more concisely highlight the key clinical paradox observed in this case—namely, the progression of PTTM despite marked tumor shrinkage following entrectinib therapy. Based on the revision discussion, we have also incorporated the hypothesis that PTTM may have developed via an atypical lymphatic route through the thoracic duct, given the absence of hematogenous metastasis. Furthermore, we noted that the tumor exhibited microsatellite stability and a low mutation burden, suggesting that these genetic alterations may not have contributed to the lymphatic spread.
To strengthen the clinical relevance, we added specific information regarding the timing of PTTM onset, which occurred approximately four weeks after the initiation of entrectinib. The short summary has also been updated accordingly.
Introduction
- Clarifying how frequently PTTM occurs in colorectal cancer (currently implied as rare, but incidence data, even from autopsy series, would help).
→ Thank you for your helpful comment regarding the incidence of PTTM in colorectal cancer. As noted in revised Introduction (page 9, lines 10-15), a large autopsy series involving 2,215 cases identified 30 instances of PTTM (1.4%), none of which were associated with colorectal cancer. This finding supports the notion that PTTM in colorectal cancer is extremely rare. The relevant section has been revised as follows: “In a large autopsy series of 2,215 cancer patients, PTTM was identified in 1.4% of cases, with gastric adenocarcinoma being the most common primary tumor [3]. Notably, none of the 30 PTTM cases involved colorectal cancer (CRC), suggesting that PTTM is exceptionally rare in CRC. However, emerging case reports indicate that it may occur in molecularly defined subtypes such as NTRK fusion-positive tumors.”
- A more explicit rationale for reporting this case (e.g., novelty due to occurrence despite tumor shrinkage with TRK inhibition).
→ Thank you for your thoughtful comment regarding the rationale for reporting this case. As suggested, we have clarified the novelty of this case in the Introduction (page 10, lines 8-15), emphasizing the clinical paradox of PTTM development despite marked tumor shrinkage following TRK inhibition. The relevant section has been revised as follows: “Specifically, we present a case of NTRK fusion-positive CRC in which PTTM developed despite a marked tumor response to entrectinib. This case offers insight into a potential disconnect between molecular tumor control and vascular disease progression. The development of PTTM, despite entrectinib’s antitumor efficacy, suggests that oncogenic signaling and pulmonary vascular pathology may proceed through particularly independent mechanisms, underscoring the need for integrated therapeutic strategies and heightened clinical vigilance.”
Case Presentation
- Suggest summarizing earlier chemotherapy regimens more concisely and focusing on changes in disease burden and symptoms that relate to PTTM risk.
→ Thank you for your constructive suggestion. In response, we have summarized the first-to third-line chemotherapy regimens more concisely and focused on the changes in disease burden and clinical symptoms relevant to PTTM risk. These revisions are reflected in the Case Presentation section (page 12, line 16 to page 14, line 9).
For example, we now state: “The initial treatment regimen consisted of capecitabine, oxaliplatin, and cetuximab. After three cycles, CT imaging revealed disease progression, with the transverse colon tumor increasing from 40 × 34 mm to 44 × 36 mm and peritoneal nodules enlarging form 44 × 32 mm to 52 × 35 mm.”
“Despite eight cycles, the disease progressed, with marked ascites accumulation and mesenteric shortening. CT at this stage showed poorly defined masses in both the transverse colon and peritoneal surfaces, without evidence of hematogenous metastases.”
“A third-line regimen of trifluridine/tipiracil (FTD/TPI) plus bevacizumab was initiated. After three cycles, ascites further increased, necessitating cell-free and concentrated ascites reinfusion therapy (CART). Throughout this period, the patient’s mild exertional dyspnea, attributable to underlying pulmonary emphysema, remained unchanged, and no respiratory symptoms suggestive o PTTM were observed.”
- Lines 116-123: The interpretation of CT findings as interstitial pneumonia vs. drug toxicity vs. infection is appropriate, but authors could indicate if bronchoalveolar lavage or other diagnosis were considered.
→ Thank you for your thoughtful comment regarding the interpretation of CT findings and the consideration of additional diagnostic procedures. In response, we have revised the Case Presentation section to clarify the differential diagnoses considered at the time of admission, including cardiogenic pulmonary edema and pulmonary embolism. However, due to the patient’s poor general condition, neither echocardiography nor contrast-enhanced CT could be performed, and a thorough evaluation was not feasible (page 15, line 11 to page 16, line 4).
We also noted that although PTTM was not initially suspected, its possibility could not be excluded at this stage (page 16, lines 4-5). Furthermore, bronchoalveolar lavage and other invasive procedures were deferred due to compromised respiratory status. These revisions are reflected in the updated Case Presentation section. (page 16, lines 5-7)
- Lines 137-143: The rationale for reintroducing entrectinib after discontinuation could be expanded—particularly given potential drug-induced lung injury.
→ The rationale for reintroducing entrectinib, which had been discontinued due to suspected drug-induced lung injury, along with the clinical findings that supported this decision at the time, has been described in the revised manuscript (page 17, line 15 to page 18, line 2). The relevant section has been revised as follows: “As pulmonary function deteriorated, entrectinib was reintroduced based on the hypothesis that TRK-driven tumor embolization within the pulmonary vasculature was contributing to disease progression. This decision was supported by the absence of hematogenous metastases and the lack of clinical improvement despite anti-infective and anti-inflammatory treatment, with no viable alternative systemic therapies available.”
Autopsy findings
- Consider including a brief table summarizing organ involvement, presence/absence of metastases, and PTTM features for quick reference.
→ Thank you for your helpful suggestion. In response, we have added a table titled “Summary of Autopsy Findings by Organ System” on page 23 to provide a concise overview of organ involvement, presence or absence of metastases, and key PTTM-related features.
- Lines 186-189: The emphasis that no hematogenous metastases were found strengthen the PTTM mechanism discussion, but authors could relate this more directly to lymphatic spread hypotheses.
→ Thank you for your insightful comment on what we consider a central aspect of this case report. In response, we have added clarifying content to the Autopsy section (page 21, lines 10-17). The relevant section has been revised as follows: “The absence of hematogenous metastases, despite extensive pulmonary vascular pathology, supports the hypothesis that PTTM may have developed through lymphatic dissemination rather than direct hematogenous spread. Furthermore, the presence of tumor emboli within lymphatic vessels across multiple organs—including the pleura, stomach, and bladder—reinforces this hypothesis. These findings suggest that lymphatic migration of tumor cells may have contributed to pulmonary microvascular obstruction and intimal proliferation, ultimately leading to the development of PTTM.”
Discussion
- Lines 230-235: The hypothesis of thoracic duct-mediated spread is interesting, this would be strengthened by citing more literature on lymphatic spread in NTRK-positive CRC.
→ Thank you for your valuable comment regarding the hypothesis of thoracic duct-mediated spread. As relevant literature on lymphatic dissemination in NTRK fusion-positive tumors remains limited, we have cited three available studies to support this discussion. These additions have been incorporated into the revised Discussion section (page 26, line 11 to page 27, line 1). Furthermore, we noted that the present case exhibited microsatellite stability and a low mutation burden, which may suggest that these genomic features are not directly involved in the lymphatic spread observed in NTRK fusion-positive colorectal cancer (page 27, lines 2-5). The relevant section has been revised as follows: “Emerging reports have described a tendency toward lymphatic spread in this molecular subtype, potentially contributing to the pathogenesis of PTTM and warranting further investigation into alternative mechanisms of tumor embolization [10]. Prior studies have documented frequent lymph node metastases in NTRK fusion-positive CRC, supporting the plausibility of lymphatic dissemination as a driver of vascular involvement [11, 12]. While this subtype is often associated with microsatellite instability and high tumor mutation burden, the present case exhibited microsatellite stability and only a modest mutation burden, suggesting that lymphatic spread and microvascular pathology may occur independently of these genomic features. These findings underscore the importance of considering noncanonical pathways of disease progression in molecularly defined CRC subtypes, particularly in the context of atypical vascular complication such as PTTM (page 26, line 14 to page 27, line 8).”
- Lines 243-249: The suggestion of combining TRK inhibitors with antiangiogenetic therapy is logical but speculative; acknowledging lack of clinical evidence in CRC-PTTM cases would provide balance.
→ Thank you for your thoughtful comment regarding the proposed combination of TRK inhibitors with antiangiogenic therapy. As you rightly pointed out, while this approach may be theoretically promising, there is currently no clinical evidence supporting its efficacy in NTRK fusion-positive colorectal cancer complicated by PTTM. To provide a balanced perspective, we have added the following statement to the Discussion section (page 28, lines 10-15): “The vascular nature of PTTM suggests that combining TRK inhibitors with antiangiogenic agents could represent a rational strategy. However, this approach remains speculative, as no clinical evidence currently supports its efficacy in NTRK fusion-positive CRC complicated by PTTM. Further investigation is warranted to determine whether vascular-targeted therapies can mitigate microangiopathic progression in this setting.”
- A brief comparison to similar published cases where tumor control did not prevent PTTM would enhance the clinical learning points.
→ Thank you for your valuable suggestion to include a comparison with similar published cases. In response, we have added a reference to a reported case of advanced non-small cell lung cancer in which fatal PTTM developed despite near-complete tumor regression following immune-chemotherapy. This addition has been incorporated into the revised Discussion section (page 27, line 17 to page 28, line 4). Specifically, we stated: “A similar paradoxical case has been reported in a patient with advanced non-small cell lung cancer who developed fatal PTTM despite near-complete tumor regression following immune-chemotherapy [10]. These observations reinforce the notion that tumor shrinkage does not necessarily equate to vascular disease control and highlight the need for adjunctive strategies targeting the pulmonary microvasculature.”
Conclusion
- Lines 252-258: The conclusion is appropriate but could be more concise, emphasizing the main takeaway which is PTTM can develop despite strong tumor responses and need parallel vascular-targeted approaches.
→ Thank you for your clear and constructive comment regarding the conclusion. In response, we have emphasized that tumor regression does not necessarily prevent the development of PTTM. To reflect this key clinical insight, we have added the following statement to the revised Conclusion section (page 29, lines 10-17): “This case demonstrates that PTTM may develop despite marked tumor regression achieved through targeted therapy, underscoring a critical dissociation between tumor control and vascular complications. Effective management may require integrated strategies that also target the pulmonary microvasculature. Further research should focus on identifying predictive biomarkers and evaluating combined approaches, including TRK inhibition and antiangiogenic therapy, to improve outcomes in patients with molecularly defined CRC, particularly those at risk for vascular complications such as PTTM.”
In addition to the reviewer’s comments, we have added a brief comparison with chronic thromboembolic pulmonary hypertension (CTEPH) to further contextualize the vascular remodeling observed in PTTM. This addition in the revised Discussion section (page 24, line 15 to page 25, line 4): “Chronic thromboembolic pulmonary hypertension (CTEPH), a sequela of unresolved embolism, may also exhibit vascular remodeling mediated by similar inflammatory and angiogenic factors—including PDGF, VEGF, and TGF-ß—suggesting partial mechanistic overlap. However, PTTM is uniquely characterized by intravascular tumor cell invasion and abrupt mediator release, resulting in rapid and aggressive microvascular obstruction. Therefore, despite overlapping molecular pathways, its clinical progression is markedly more aggressive than that of CTEPH [3, 9].”
We sincerely thank you for taking the time to review our manuscript despite your busy schedule. Your clear and insightful comments have greatly contributed to the refinement of our work. Through the process of revising the manuscript in accordance with your suggestions, we gained new perspectives and a deeper understanding of the case, which we believe has led to a clearer and more meaningful presentation. We are truly grateful for your thoughtful guidance.
I hope these revisions meet your requirements.
Sincerely,
Hideki Nagano, MD, PhD
Department of Surgery, Marunouchi Hospital, Japan
Reviewer 2 Report
Comments and Suggestions for Authors
This is a well-written and informative case report describing a rare but clinically relevant event: the development of pulmonary tumor thrombotic microangiopathy (PTTM) in a patient with NTRK fusion-positive colorectal cancer during effective treatment with entrectinib. The authors present detailed clinical, radiological, and pathological findings, and the discussion appropriately highlights the diagnostic challenges, underlying mechanisms, and therapeutic implications. The case adds value to the literature, given the scarcity of reports addressing the coexistence of NTRK-driven CRC, TRK inhibition, and PTTM.
While the case is rare, the discussion could be enriched by explicitly stating how this report differs from prior descriptions of PTTM in gastrointestinal malignancies. Clarifying whether this is the first reported case of PTTM in NTRK fusion-positive CRC treated with entrectinib would strengthen the novelty claim.
Author Response
We sincerely appreciate your thoughtful and constructive comments on our manuscript. In accordance with your suggestions, we have revised the text to address each point raised. Below, we provide detailed explanations of the changes made in response to your feedback. Page and line numbers refer to the revised manuscript with tracked changes, and the modified text is highlighted in red clarity.
While the case is rare, the discussion could be enriched by explicitly stating how this report differs from prior descriptions of PTTM in gastrointestinal malignancies.
Clarifying whether this is the first reported case of PTTM in NTRK fusion-positive CRC treated with entrectinib would strengthen the novelty claim.
→ Thank you for your insightful comment regarding the novelty and comparative content of this case. In response, we have clarified how this report differs from prior description of PTTM in gastrointestinal malignancies. Specifically, this case highlights the unexpected development of PTTM despite a strong antitumor response to TRK inhibitor therapy, underscoring that tumor regression does not necessarily prevent vascular complications. Additionally, we discuss the possibility of thoracic duct-mediated lymphatic spread in NTRK fusion-positive colorectal cancer, which may differ from conventional metastatic pathways. The tumor’s microsatellite stability and low mutation burden further suggest that these genomic features may not directly contribute to the observed lymphatic dissemination.
To emphasize the novelty of this case, we have explicitly stated that this is the first reported instance of PTTM occurring in NTRK fusion-positive colorectal cancer treated with entrectinib in Introduction section (page 9, lines 15-16): “To our knowledge, no prior reports have described PTTM in NTRK fusion-positive CRC.”
I hope these revisions meet your requirements.
Sincerely,
Hideki Nagano, MD, PhD
Department of Surgery, Marunouchi Hospital, Japan